# Clinical Scores, Biomarkers and IT Tools in Lung Cancer Screening—Can an Integrated Approach Overcome Current Challenges?

**DOI:** 10.3390/cancers15041218

**Published:** 2023-02-14

**Authors:** Wieland Voigt, Helmut Prosch, Mario Silva

**Affiliations:** 1Medical Innovation and Management, Steinbeis University Berlin, Ernst-Augustin-Strasse 15, 12489 Berlin, Germany; 2Department of Biomedical Imaging and Image-Guided Therapy, Medical University of Vienna, Vienna, General Hospital, 1090 Vienna, Austria; 3Scienze Radiologiche, Department of Medicine and Surgery (DiMeC), University of Parma, 43121 Parma, Italy

**Keywords:** lung cancer screening, clinical scores, biomarker, computed tomography, pulmonary nodule

## Abstract

**Simple Summary:**

There are more chances to successfully treat lung cancer if the disease is detected early. Screening for lung cancer with low dose computed tomography in people at higher risk is a powerful tool for early lung cancer detection. However, several factors need to be considered for the selection of candidates who might benefit most from screening. Then, the process of low dose computed tomography needs to keep up with technical advances that offer a higher precision in the detection of potential lung cancer nodules. If nodules are detected, additional data might help physicians decide whether they are benign or malignant and determine the appropriate further procedure. In this review, we describe current limitations and advances of these different aspects of lung cancer screening. Further research is required but the integration of scientific and technological progress might improve the performance of lung cancer screening generally.

**Abstract:**

As most lung cancer (LC) cases are still detected at advanced and incurable stages, there are increasing efforts to foster detection at earlier stages by low dose computed tomography (LDCT) based LC screening. In this scoping review, we describe current advances in candidate selection for screening (selection phase), technical aspects (screening), and probability evaluation of malignancy of CT-detected pulmonary nodules (PN management). Literature was non-systematically assessed and reviewed for suitability by the authors. For the selection phase, we describe current eligibility criteria for screening, along with their limitations and potential refinements through advanced clinical scores and biomarker assessments. For LC screening, we discuss how the accuracy of computerized tomography (CT) scan reading might be augmented by IT tools, helping radiologists to cope with increasing workloads. For PN management, we evaluate the precision of follow-up scans by semi-automatic volume measurements of CT-detected PN. Moreover, we present an integrative approach to evaluate the probability of PN malignancy to enable safe decisions on further management. As a clear limitation, additional validation studies are required for most innovative diagnostic approaches presented in this article, but the integration of clinical risk models, current imaging techniques, and advancing biomarker research has the potential to improve the LC screening performance generally.

## 1. Introduction

Lung cancer (LC) is the leading cause of cancer related mortality because most patients are diagnosed at late stages when prognosis is poor. This fact is unsatisfactory, given that recent LC screening trials have demonstrated that annual low dose computed tomography (LDCT) screening can reduce LC mortality in high-risk populations such as heavy smokers [1,2,3]. Despite the positive results of screening trials, nationwide screening programs are only implemented in few countries. However, the recruitment and engagement of screening candidates remains low and unsatisfactory in countries with established screening programs, such as the United States [4,5].

An important reason for the poor implementation of screening programs might be found in the imperfect criteria for the selection of LC screening candidates that might exclude various risk groups. For instance, the National Lung Screening Trial (NLST) enrollment criteria primarily focus on age and smoking history [6]. In fact, it has been estimated that applying these criteria would miss more than 50% of incident LC cases [7,8], among which at least 25% are not due to smoking [9]. The United States Preventive Service Task Force (USPSTF) is endorsing more inclusive selection criteria, currently set at 20 instead of 30 pack-years and 50 instead of 55 years of age [10]. However, LC high-risk factors obviously differ between regions, as in the China population, more than 90% of LC cases were outside the current screening criteria [11].

Other drawbacks of current LC screening strategies are the high rate of false positive results [1,3,12] and the high prevalence of indeterminate nodules, leading to follow-up diagnostic procedures that are associated with increased radiation exposure, overdiagnosis, and anxiety [13,14]. Increased specificity in LC screening, for example through the integration of automated volumetric nodule assessment as well as imaging- and blood-based biomarkers, would improve nodule management. Dedicated IT solutions might additionally facilitate the identification and management of incidental non-cancerous findings during LDCT, which has been shown to improve patient health by enabling earlier treatment of undiagnosed cardiovascular or respiratory disease [15,16,17].

Compared to the NLST criteria, the Lung Report And Data System (Lung-RADS) classification system for LDCT-based LC screening [18,19] reduces the false positive rate by both increasing the size threshold of pulmonary nodules (PN) at baseline and requiring growth for preexisting PN at follow-up scans after 3 or 6 months, thereby influencing screening sensitivity [20,21]. Several biomarkers and/or radiomics-based risk assessments have been recently suggested to further support PN interpretation, to determine appropriate intervals for computerized tomography (CT) follow-up scans, and to support clinical decisions on further PN management such as biopsy or surgery [13,21,22,23].

In this scoping review, we discuss current developments regarding the selection criteria for LC screening (selection phase), the potential improvements of LC screening through computerized image assessments and nodule interpretation (screening), and the potential of clinical scores and biomarkers for further risk assessments (management).

## 2. Aims of This Study and Methods

Given the variety of issues we intended to cover in this scoping review, we decided to follow a non-systematic approach to identify most recent literature on potential improvements of (1) selection criteria for LC screening, (2) nodule characterization during LC screening, and (3) lung nodule management and risk prediction following LDCT scans. At first, we collected most recent key original and review articles by searching PubMed and Google Scholar databases (dates searched ranged from 2011 to 2022) as well as reference lists of the retrieved articles. The obtained literature was then reviewed by the authors to identify key findings and categorized into the sections ‘selection phase’, ‘screening’, and ‘management’ according to their general relevance. Search terms queried in title, abstract, and keywords included “lung cancer screening” or “low dose computed tomography” or “clinical scores” or “patient selection OR risk stratification” or “risk models” or “biomarkers” or “incidental findings” or “(lung OR pulmonary nodule) AND management” or “lung cancer risk prediction”. Filters and additional search terms such as “ctDNA OR cfDNA” or “radiomics” were used to narrow search results.

Articles were assessed by all authors to identify LC screening relevant approaches in or near clinical application. Articles on approaches that are still in a rather experimental and developmental stage were excluded, unless groundbreaking results were presented. On this basis, an integrated approach consisting of clinical risk factors and biomarkers to improve risk stratification before and after LDCT screening as well as IT tools to increase the diagnostic and prognostic yield of CT-imaging was developed. The remainder of this review is divided into the sections (1) selection phase, (2) screening, and (3) management (Figure 1).

## 3. Selection Phase: Criteria for Lung Cancer Screening

The definition of appropriate selection criteria is of paramount importance for any cancer screening program. Besides well-established criteria which primarily focus on smoking history and age, more sophisticated selection criteria that include the presence of specific biomarkers could further improve the efficacy of LC screening (Figure 1).

### 3.1. Risk Factors and Risk Models

Simulation models demonstrated that the refinement of eligibility criteria for LC screening could capture more LC cases not meeting current eligibility criteria. For the USPSTF 2013 criteria basing on age and smoking history, for example, the reduction of both the lower age limit from 55 to 50 years and the minimum smoking history from 30 to 20 pack-years increased both the screening eligibility, including that for different ethnicities and high-risk women, and the number of LC deaths prevented [6,24]. In 2020, draft recommendations were issued by the USPSTF to screen younger patients with less smoking history and to include more racial and ethnic minorities [25]; however, Lozier et al. suggested that social determinants of healthcare need to be additionally considered to avoid racial and ethnic disparities [26].

Risk models, however, apply different predictors such as additional smoking exposure variables, education, body mass index, chronic obstructive pulmonary disease (COPD), history of cancer, and ethnicity. The Prostate, Lung, Colorectal, and Ovarian (PLCO) Cancer Screening Trial (PLCO_M2012_), for example, demonstrated a significantly higher sensitivity, predictive value, and cancer detection rate than the age and smoking history-based eligibility criteria of the NLST (*p* = 0.009) or the Nederlands–Leuvens Longkanker Screenings Onderzoek Trial (NELSON; *p* = 0.003) [27]. Furthermore, PLCO_M2012_ was reported to outperform USPSTF2013 criteria in efficiently selecting individuals for LC screening [28].

Ethnicity is paramount when dealing with LC epidemiology. Risk models applying other or additional predictors showed good discriminative power for the selection of never-smoking females in Asia affected by slow-growing adenocarcinoma [29], which is of particular importance as about one quarter of LC cases arise in never-smokers [30]. The rising incidence of LC amongst non-smokers can be attributed to environmental and occupational exposure to various kinds of hazardous substances, such as asbestos, ionizing radiation, vinyl chloride, outdoor air pollution, second-hand and indoor smoke, arsenic, beryllium, chromium, and nickel [31,32]. Accordingly, novel risk models such as the Liverpool lung project risk model [33] comprise the predictors age, sex, smoking status and duration, asbestos exposure, and non-cancer lung disease.

Obviously, more advanced models applying risk factors beyond smoking and age demonstrated better effectiveness in the selection of participants for LDCT screening. Because altered molecular features derived from LC cells and the tumor microenvironment can be detected and quantified by sensitive technologies during early carcinogenesis, additional risk stratification by use of biomarkers has been suggested to further improve the selection of suitable patients for LDCT screening, and a number of biomarkers have been extensively investigated over the last few years.

### 3.2. Biomarkers

#### 3.2.1. Protein Panels and Autoantibodies

In a validation study on 63 ever-smoking LC patients and 90 matched controls, a risk score based on four circulating protein biomarkers (cancer antigen 125 [CA125], carcinoembryonic antigen [CEA], cytokeratin-19 fragment [CYFRA 21-1], precursor form of surfactant protein B [pro-SFTB]) considerably improved the USPSTF eligibility criteria (area under the curve [AUC] 0.83 [95% confidence interval (CI), 0.76–0.90] vs. AUC 0.73 [95% CI, 0.64–0.82]; *p* = 0.003) and 1-year LC prediction [34]. Likewise, the EarlyCDT-Lung test (EarlyCDT-Lung, Oncimmune Ltd., Nottingham, UK [35]) assesses the presence of seven cancer associated autoantigens (p53, NY-ESO-1, CAGE, GBU4-5, HuD, MAGE A4, SOX2) followed by LDCT scanning 6-monthly in case of positive results. This combination showed high specificity (90.3% [95% CI, 89.5–91.0]) and resulted in a high detection rate of stage I/II LC cases in adults at increased LC risk, as defined by age, smoking history, and family history of LC (positive predictive value 1.2% [95% CI, 0.5–2.4]; negative predictive value: 100.0% [95% CI, 99.9–100.0]). These findings demonstrated the value of the EarlyCDT-Lung test as selection phase biomarker to improve eligibility criteria. Another approach combined a panel of proteins (CEA, CYFRA 21-1, CA125, hepatocyte growth factor) with the New York esophageal cancer-1 antibody and demonstrated that the accuracy of age and smoking history-based selection (AUC 0.68) can be increased by the combination with these biomarker variables (AUC 0.86; biomarker alone: AUC 0.81 [36]).

#### 3.2.2. Cell-Free DNA and DNA Methylation

The machine learning-based Lung Cancer Likelihood in Plasma (Lung-CLiP) approach comprises the targeted sequencing of plasma-derived cell-free DNA (cfDNA) and analysis of single nucleotide variants (SNV) as well as genome-wide copy numbers to provide a likelihood score for the presence of LC-derived cfDNA in blood samples [37]. At 98% (80%) specificity, sensitivities of 41% (63%) at stage I, 54% (69%) at stage II, and 67% (75%) at stage III were observed [37]. Another approach comparing the genome-wide fragmentation patterns of cfDNA demonstrated high sensitivities for the detection of different cancers (57% to >99%) at 98% specificity, with an overall AUC of 0.94 [38]. Likewise, PanSeer, a noninvasive blood test assessing methylation of circulating tumor DNA (ctDNA) detects cancer in 95% (95% CI, 89–98) of asymptomatic patients who were later diagnosed [39], and analysis of SHOX2 and PTGER4 methylation in plasma DNA allowed significant differentiation of LC patients from individuals without malignancy (AUC 0.88; sensitivity at 90% specificity: 67% [40]).

#### 3.2.3. miRNA

MicroRNAs (miRNA) are noncoding and stable RNA fragments regulating gene expression post-transcriptionally. A meta-analysis on 65 LC publications (6919 LC patients and 7064 controls) showed that miRNA derived from circulating tumor cells (CTC) can be detected with a sensitivity of 0.83 and a specificity of 0.84 (AUC 0.90 [41]). In a recent study, a 14-miRNA set distinguished early-stage LC patients with symptoms from individuals without LC, with an accuracy of 95.9% (95% CI, 95.7–96.2), sensitivity of 76.3% (95% CI, 74.5–78.0), and specificity of 97.5% (95% CI, 97.2–97.7 [42]).

#### 3.2.4. Other Biomarkers

There are additional biomarkers under development that are potentially useful for LC screening. One interesting approach might be the detection of volatile organic compounds (VOCs) in exhaled breath [43,44,45,46]. In a small study on a potential breath test detection model that was built on exhaled breath samples from 139 LC patients and 289 controls, the validation set comprising 47 participants revealed a sensitivity of 100%, a specificity of 92.86%, and an accuracy of 95.74% (AUC 0.9586 [45]). Another study demonstrated that the combination of clinical parameters and exhaled-breath data in an artificial neural network resulted in good performance (AUC 0.84; 95% CI, 0.79–0.89) and might therefore enhance risk stratification in LC screening [46].

### 3.3. Summary Selection Phase

The spectrum ranging from autoantibodies and protein panels to SNV, ctDNA methylation, cfDNA, miRNA, and other potential biomarkers demonstrated promising results in several studies. However, none of these biomarkers seems sufficiently validated for clinical routine use, and further large-scale clinical studies in true screening settings are required to generate proper evidence. Once clinically validated, these biomarkers could play an important role for a more refined selection of individuals for LC screening.

Lung cancer screening reduces LC-specific and all-cause mortality. However, while narrow selection criteria basing on age and smoking history miss a significant number of LC cases, the widening of current selection criteria increases false positive rates. Advanced models including risk factors beyond smoking and age demonstrated increased effectiveness. As an important step further, a combination of advanced risk models comprising clinical, occupational, and environmental factors are awaited along with validated LC biomarkers to help increasing the pre-test probability and reducing the false positive rate in LDCT screening. As high false positive rates might contribute to low participation in LC screening, an improved risk stratification might even increase the acceptance rate of screening programs.

## 4. Screening: Computer-Aided Detection and Radiomics

In LDCT screening, the vast majority of individuals shows PN of any size [47], but only 3.6% of the detected PN are later diagnosed as LC [1], underlining the importance of specific PN characterization during LC screening. To support the actual LDCT screening process, computer-aided detection (CAD) has been increasingly employed for automatic identification of PN, as a complementary tool to visual reading. The CAD systems not only provide a second opinion for image interpretation, but also contribute to reduced false-negative rates [48] and decreased inter-observer variation [49]. In addition, CAD systems accelerate the screening workflow and support lung nodule management [50,51,52].

Most importantly, CAD systems combined with deep learning are gaining momentum for automatic stratification of nodule malignancy likelihood [53]. Basically, a CAD system comprise components for data acquisition and pre-processing, lung segmentation, PN detection, as well as PN segmentation and characterization [53]. Deep learning algorithms are increasingly employed for lung segmentation, utilizing three-dimensional lung segmentation improved by the adversarial neural network training [54], as well as PN detection [55,56]. As PN detection still results in a considerable number of false-positive candidates, several approaches for PN feature extraction and classification have been applied for the reduction of false positive rates [57,58]. Despite promising developments, existing CAD algorithms for LC diagnosis still require further improvements because a strictly defined set of features differentiating between benign and cancerous PN is still missing.

Radiomics, as a further development of the CAD approach, is based on the extraction of a large number of such medical image features that support the identification of cancer characteristics using data-characterization algorithms. As radiomics is still far from clinical standardization and use, we present only few studies providing promising results on malignant-benign differentiation, staging, and PN classification. Kumar et al. [59] reported an accuracy of 79.06%, a sensitivity of 78.00%, and a specificity of 76.11%. Another radiomic study by Liu et al. [60] demonstrated an accuracy of 81%, a sensitivity of 76.2%, and a specificity of 91.7%. In addition, radiomic features have been shown to contribute to tumor staging [61,62], and radiomic features might aid to improve the classification of PN into high- and low-risk PN, thereby reducing the rate of indeterminate PN [63]. The reference standard for PN characterization is currently based on clinical and evolutional characteristics that have been recently applied in a deep learning approach combining three-dimensional CT scans, physiological symptoms, and clinical biomarkers, providing sensitivity and specificity values of 94% and 91%, respectively [64]. A deep learning algorithm using the patient’s current and prior CT volumes has been demonstrated to specifically predict LC risk (AUC: 94.4%) [65].

The LC screening programs further offer the opportunity to incidentally identify individuals with undiagnosed cardiovascular and respiratory disease [66,67,68]. The NLST data revealed that emphysema was detected in 44.2% of 25,002 participants who had undergone LC screening including baseline and follow-up scans, while history of COPD/emphysema was reported in only 10.6%. Emphysema found by LDCT screening was associated with a significantly increased respiratory disease mortality hazard ratio (2.27; 95% CI, 1.92–2.7) [68]. Emphysema increases LC risk, and airflow obstruction has recently been shown to be an independent risk factor for LC risk at baseline LDCT [69]. In another study, coronary artery calcification was found in 61.9% of 680 individuals who had undergone LDCT screening, demonstrating the additional benefit offered by LC screening to detect cardiovascular disease [67]. Most of reported incidental findings require follow up imaging and further diagnostics including sometimes even invasive procedures to confirm or rule out underlying diseases, mainly other types of cancer [70,71]. Newly diagnosed non lung cancer conditions resulted in management changes like alteration in medications in a relevant subset of patients [70,71]

## 5. Management: Pulmonary Nodules and Risk Prediction

### 5.1. Clinical Scores

After LDCT, the decisive criterion for further PN management is the probability that the nodule is malignant (Figure 2). High probability of malignancy requires a more aggressive assessment or surgical resection, whereas intermediate and low probabilities of malignancy demand further evaluation by tissue biopsy or short- and regular-interval surveillance, respectively [72]. The importance of accurate probability estimates lies in the facts that, on the one hand, most incidental or screen-detected PN are benign, and, on the other hand, mortality dramatically increases with higher tumor stages at diagnosis [73].

Validated probability models combine clinical characteristics with PN imaging features that have been shown to be independent LC predictors [74,75,76,77,78,79,80,81,82]. Most models include patient characteristics such as age, smoking history, prior malignancy, as well as PN characteristics such as location, edge characteristics, size, and growth [74,75,76,80], while other models add the results of fluorodeoxyglucose-positron emission tomographic (FDG-PET) scans [77], symptoms such as hemoptysis [74,75,81], and the presence of spiculation [76,79,80,81]. Fair to good sensitivities and specificities have been reported, with AUC between 0.79 and 0.92. However, a comparison of models estimating the probability of PN malignancy in defined clinical scenarios demonstrated that the accuracy of these models is highest in populations similar to those in which they were developed [83]. Models derived from high-risk populations with a higher prevalence of malignancy tended to overestimate the probability of malignancy of PN proven to be benign and vice versa. Moreover, most models were developed based on relatively homogeneous populations, and ethnicity has not been evaluated as a predictor of malignancy. In some studies, expert physician assessment performed equal or better than probability models [84,85,86]. Subsolid PN remain a topic of scientific debate, as they have a higher risk of malignancy than solid PN but exhibit a more indolent behavior.

Recent advances in imaging and molecular research have identified an increasing number of radiomic features as well as biomarkers indicative of malignancy. As described below, the combination of clinical predictors with radiomic features and molecular biomarkers may result in accuracies superior to those obtained by clinical models alone (Figure 2).

### 5.2. Volumetry

Current data suggest that volumetric tumor measurements during LDCT improve the decision making for individual patients. In the NELSON-trial [2], volume CT screening of high-risk participants reduced LC mortality after 10 years of follow-up when compared to no screening (cumulative rate ratio for LC-related death 0.76 [95% CI, 0.61–0.94; *p* = 0.01]). Importantly, volume CT screening assessing the volume doubling time for the follow-up of indeterminate PN significantly reduced false positive results and unnecessary procedures [2,87].

Semi-automatic volume measurement of solid PN detected during LDCT screening clearly outperformed manual diameter measurements [88,89]. The systematic error occurring during manual diameter measurement exceeded the cut-off values indicative for nodule growth, potentially resulting in misinterpretation and substantial misclassification. This effect, in contrast, was almost absent in semi-automated volume measurements, which is of utmost significance for the follow-up management of patients in LC screening programs [89]. It is important to note, however, that the appropriate nodule size threshold for recall at baseline LC screening depends on the nodule volumetry software used [90]. Artificial intelligence as a standalone reader to automatically detect and classify solid PN reduced the rate of negative misclassifications as well as the radiologists’ workload at LDCT baseline screening [91]. Automation of volumetric nodule classification might not only contribute to a better assessment of the dynamics of nodule development (stable vs. growing), but also reduces the radiologists’ time requirements during follow-up LDCT [88,92].

### 5.3. Radiomics and Artificial Intelligence Applications

To further improve PN management in the LDCT screening setting, the radiomic approach contributes to the enhancement of risk prediction models by the assessment of additional CT image features (Figure 2). A multiparameter model involving nodule size, CT parameters, and radiomorphologic nodule characteristics has been shown to more accurately discriminate adenocarcinoma from minimally invasive and in situ adenocarcinoma in lung pure ground-glass nodules [22]. Likewise, the Pan-Canadian Early Detection of Lung Cancer Study models basing on participants’ characteristics and LDCT imaging parameters demonstrated very good discrimination in the prevalence screening setting and has the potential to improve PN management, including decision-making on further procedures required such as biopsy and short-term follow-up scan [93]. Automated PN detection on LDCT scans using a convolutional neural network-based prototype demonstrating high sensitivity and specificity for PN and coronary artery calcium volume improved the prediction of LC and cardiac events at the 1-year follow-up [16].

Risk prediction has made further progress with the implementation of deep machine learning. Huang et al. [94] reported on the development of a deep machine learning algorithm recognizing temporal and spatial changes of PN related and non-PN related features in CT scans and combining these data with clinical information. The algorithm demonstrated an excellent discrimination at 1-, 2-, and 3-year follow-up, with AUC values for LC diagnosis of 0.968 ± 0.013, 0.946 ± 0.013, and 0.899 ± 0.017, respectively. Radiomic models therefore allow a more accurate classification of high- and low-risk patients than Lung-RADS [94,95] and nodule volume-doubling time [94]. Deep learning image reconstruction has been shown to decrease image noise and to improve both PN detection rate and measurement accuracy on ultra-LDCT images [96]. In current LC screening programs, the timing of follow-up CT-scans is determined based on mean PN diameter, volume or density of the largest PN, and the occurrence of new PN. The radiomic and deep machine learning approach might add even more accuracy for risk prediction and the timing of follow-up scans (Figure 2).

### 5.4. Biomarkers

Molecular biomarkers analysis might be a meaningful adjunct approach to current risk classification by follow-up scans of indeterminate PN. Computerized tomography usually identifies an excessive number of indeterminate PN, and even though most of these nodules are benign, many patients undergo unnecessary procedures such as lung biopsy and overtreatment. A validated biomarker approach might refine current risk classification, especially if the probability of malignancy is in the intermediate range, thereby limiting the number of false positives and improving the identification of early-stage LC.

#### 5.4.1. Proteins and Autoantibodies

In a prospective observational trial on 685 patients with PN 8 to 30 mm in diameter, the relative abundance of two plasma proteins (LG3BP, C163A) as measured by multiple reaction monitoring mass spectroscopy was integrated in a risk prediction model to distinguish benign from malignant PN. With a sensitivity of 97% (95% CI, 82–100), a specificity of 44% (95% CI, 36–52), and a negative predictive value of 98% (95% CI, 92–100), this classifier would reduce the procedures performed on benign nodules by 40% [97]. Likewise, Trivedi et al. reported on a support vector machine learning algorithm combining the results of a plasma-based multiplexed protein assay with clinical factors. This model demonstrated a negative predictive value of 94% (sensitivity 94%) and, therefore, might serve as a rule-out test for patients with benign disease to avoid unnecessary interventions [97]. The ELISA-based detection of complement C4d in plasma samples improved the risk classification of indeterminate PN but could not discriminate between asymptomatic high-risk individuals with or without early LC [98]. Measurement of complement C4c, cytokeratin fragment 21-1 (CYFRA 21-1), and C-reactive protein (CRP) then discriminates between benign and malignant PN specifically (AUC: 0.86; 95% CI, 0.80–0.92; specificity: 92%), and, in combination with clinical factors, might contribute to a reclassification of intermediate-risk patients with indeterminate PN into patients requiring a more aggressive workup [99]. A four-marker protein panel (pro-SFTPB, CA125, CYFRA 21-1, CEA) clearly outperformed nodule size-based risk classification by increasing sensitivity at high specificity. In particular, the performance of the protein panel combined with PN size was especially relevant for individuals with nodule sizes of less than or equal to 6 mm (AUC: 0.95; 95% CI, 0.85–1.00) [100]. Likewise, a model combining serum biomarkers (ProGRP, CEA, SCC, CYFRA21-1), clinical information, risk factors, and LDCT results demonstrated a significantly higher AUC (0.9151 vs 0.8360; *p* = 0.001) than the American College of Chest Physicians model, a nodule size-based model [101].

In a prospective registry study, the predictive value of nodule size-based risk assessments could also be increased by measuring autoantibodies to seven tumor-associated antigens (EarlyCDT-Lung). In case of a positive antibody test, the risk for LC development increased 2.7-fold for PN smaller than 20 mm in diameter, which would support further management of relatively small and indeterminate PN and allow early LC detection [102]. In a retrospective assessment on 397 patients with pulmonary lesions and 74 controls, a set of seven tumor-associated autoantibodies combined with CT could identify malignant PN less than 8 mm in diameter, with a specificity of 95.8% [103]. Another study applying the EarlyCDT-Lung test demonstrated a potential shift to localized stage diagnosis in 10.8% of indeterminate PN leading to more patient lives saved [104].

#### 5.4.2. Cell-Free DNA and DNA methylation

The highly sensitive detection of LC-specific changes of DNA methylation in plasma-derived cfDNA samples could be demonstrated to identify high-risk patients and to improve early LC diagnosis by differentiating malignant from benign nodules in CT-detected PN [38,105,106,107,108]. In one study, a three-gene combination (CDO1, SOX17, HOXA7) yielded a specificity and sensitivity of 90% and 71%, respectively, and the combination with clinical predictors further improved the diagnostic accuracy of the test from AUC 0.88 (95% CI, 0.84–0.93) to AUC 0.94 (95% CI, 0.91–0.96) [105]. In a similar study, two three-gene combinations detected LC-specific DNA methylation in sputum (TAC1, HOXA-7, SOX17) and plasma (CDO1, TAC1, SOX17), with a diagnostic accuracy of AUC 0.89 (95% CI, 0.80–0.98) and AUC 0.77 (95% CI, 0.68–0.86), respectively [106]. In particular, the assessment of plasma cfDNA achieves promising sensitivity in very early-stage LC [108] and allows the differentiation from tuberculosis [107], both contributing to an improved management of indeterminate PN. The detection of specific fragmentation patterns of blood-derived cfDNA combined with machine learning was additionally able to determine the tissue of origin of cancer, which might be of high relevance to asses if a malignant PN originated from primary LC or metastasis of other tissues [38].

Likewise, the assessment of blood-derived DNA methylation biomarkers (PTGER4, RASSF1A, SHOX2) combined with radiological characteristics (PN diameter) demonstrated a promising predictive performance for malignancy (AUC 0.951) among individuals with CT-detected PN (nodule size between 11.22 ± 7.56 [benign] and 21.83 ± 10.88 [malignant]) [109]. The analysis of RUNX3 and RASSF1A promoter methylation on biopsy and serum-derived samples might be another interesting approach to distinguish between benign and malignant PN, as solitary PN ≤10 mm in size were included in this study [110]. In addition to DNA methylation analysis, cfDNA obviously offers additional examination possibilities such as the detection of driver mutations and whole exome sequencing [111], underlining the diagnostic capability of cfDNA to discriminate malignant from benign nodules.

#### 5.4.3. miRNA

Recent studies demonstrated that the combination of CT data with the results of miRNA analysis could improve LC diagnosis and management [112,113,114]. In one study, a panel of five miRNA exhibited low sensitivity for the detection of different LC pathologies (overall 34.0%) in 369 individuals with PN detected by CT. However, combining miRNA test positivity with CT imaging reduced the false positive rate for nodules and glass ground nodules from 33.1% to 3.2% [112]. Likewise, the combination of two miRNA biomarkers with nodule diameter on CT images improved LC diagnosis among indeterminate PN, thereby helping to avoid unnecessary biopsies, follow-up CT, and anxiety of patients [114]. Furthermore, the use of miRNA was prospectively investigated for the allocation of patients to specific LDCT screening intervals. Pastorino et al. [113] showed in the BioMILD trial that miRNA profiling at the time of baseline LDCT-scan improves the individual risk prediction, especially in individuals with baseline indeterminate or positive LDCT results, thereby providing the opportunity both to guide subsequent diagnostic procedures and to personalize LDCT screening intervals. Participants with negative miRNA signature classifier and negative LDCT, representing 64.7% of a population selected by age (50–75 years) and smoking history (≥30 pack-years), were assigned to LDCT screening every 3 years, resulting in a LC incidence as low as 0.8% at 4 years [113].

Several models combining the expression of two to three miRNA specimens with CT imaging features [115,116] and, additionally, protein antigens [117] demonstrated moderate discriminatory accuracy to predict malignant nodules. Due to the retrospective study design and limited samples size, these models, despite promising results, might be considered as being in a rather developmental state. Kossenkov et al. [118], however, reported on the development of a PN classifier basing on 41 RNA biomarkers that outperformed clinical algorithms in discriminating malignant from benign PN (6–20 mm) and could therefore contribute to an improved decision making in the workup for indeterminate PN.

#### 5.4.4. Circulating Tumor Cells

As circulating tumor cells (CTCs) are tumor cells shedding from either primary tumors or its metastases, CTC detection by the noninvasive liquid biopsy approach has shown promises in cancer diagnosis, prognosis, and prediction [119]. Several CTC detection approaches utilize the folate receptor (FR) for CTC labeling because FR is highly upregulated in non-small cell LC, and only a few FR expressing cells are present in the peripheral blood, including CTC and a rare subtype of monocytes [120].

Xue et al. [121] demonstrated the clinical relevance of the detection of FR-positive CTC as a companion assay for LC screening, suitable for early diagnosis of patients with CT-detected small PN. In their study on 72 LC patients and 26 controls, the assay achieved AUC, sensitivity, and specificity of 0.8063 (95% CI, 0.6769–0.9356), 80.00%, and 75.00%, respectively, if nodule size was equal to or below 30 mm. This finding suggests considerable discriminatory potential of this approach for small indeterminate PN identified by LDCT.

Moreover, FR-positive CTC count combined with the maximum tumor diameter was shown to differentiate non-invasive from invasive cancers (sensitivity 63.6%–81.8%; specificity of 71.4%–89.7%) in 382 patients with suspicious PN on CT [122]. As PN malignancy obviously correlates with CTC levels in the peripheral blood [123,124,125,126], this approach might be a valuable tool to inform nodule management after LDCT scans. Importantly, most studies demonstrated that models combining CTC levels with additional biomarkers or nodule characteristics perform better than single parameter models [122,123,126].

#### 5.4.5. Metabolites

Metabolic processes including fat-, protein-, and sugar metabolism are altered during tumorigenesis, a fact that is mirrored by several molecular features detectable in biofluids such as serum [127,128,129], urine [128], and saliva [130]. A number of recent experimental studies reported on metabolic signatures that separates LC cases from healthy controls and therefore offer potential clinical applications. In a pilot study on 31 LC patients and 92 matched healthy controls, a metabolic signature of nine metabolites identified by gas chromatography coupled to mass spectrometry (GC/MS) allowed to discriminate cancer and healthy samples with 100% sensitivity and 95% specificity (AUC 0.99 [127]). In a single cohort of 35 LC patients, 48 metabolic changes could be identified by nuclear magnetic resonance and mass spectrometry in urine and blood samples obtained before and after surgical tumor resection [128]. Laser desorption/ionization (LDI) mass spectrometry-based liquid biopsy on serum samples of 233 healthy controls and 950 patients with different cancers, combined with machine learning for high-throughput analysis, identified 10 discriminative features for each cancer (AUC 0.922 for non–small-cell LC [NSCLC]; [129]). Likewise, a multiple logistic regression model derived from the profiles of 10 salivary metabolites identified in 41 LC patients and 21 patients with benign nodules could clearly discriminate LC from benign lesions (AUC 0.729 [130]). These results might require appropriate validation in clinical trials, but metabolite profiling might support PN management after indeterminate CT findings.

### 5.5. Summary Management

Radiomics as well as liquid biopsy biomarkers such as cfDNA, ctDNA, miRNA, exosomes, and CTC have shown promise to differentiate malignant from benign PN for early diagnosis, risk evaluation, and decision on tailored diagnostic and therapeutic procedures (Figure 2). As non-invasive or nearly non-invasive approaches, radiomics as well as biomarkers are applicable in patients not eligible for tissue biopsy and allow serial measurements, thereby avoiding sampling bias. However, large clinical studies are still required to assess the clinical utility of most assays [131,132,133,134]. In this regard, a recent review reported on well-validated liquid biopsy biomarkers and proposed an interesting strategy combining LDCT scans and biomarkers for early LC characterization [135].

## 6. Discussion

In this scoping review, we describe a potential three-step strategy to overcome current limitations in LC screening (Figure 1). First, our approach proposes improved selection criteria and strategies for LC screening programs both to broaden the eligible population and to increase the pre-test probability to guide selection. Second, technical progress may advance the actual LDCT screening process to a new level by the automation of nodule identification and reduction of false positives. Moreover, state-of-the-art LC screening offers additional diagnostic opportunities such as the incidental identification of patients with undiagnosed cardiovascular and respiratory disease. Third, elaborated criteria to evaluate the probability of PN malignancy would allow to make safer decisions on further PN management, including personalized screening intervals, tissue biopsy, and surgical resection.

Such a strategy obviously needs to be based on proper validation studies providing sufficient evidence for each element. There may be plenty of evidence demonstrating the efficacy of single elements such as risk scores for the selection of LC screening candidates [10,27,33], volumetric tumor measurements during LDCT [2,87], and clinical scores evaluating the probability of malignancy of LDCT-detected PN [74,75,76,77,78,79,80,81,82]. However, the strength of evidence varies widely among studies on biomarker development and performance, ranging from small retrospective studies to large prospective trials [2,35,39,101,113,136]. For most diagnostic approaches discussed in this article, additional evidence needs to be developed by validation studies to transfer promising approaches from basic research into clinical application. As pointed out before, many risk scores and probability models are not globally applicable because risk factors differ among populations [26,29,30,83]. The same is probably true for many biomarker assessments as gene expression patterns differ among ethnic groups [137,138]. These facts must be considered in future validation studies. Aside from the challenges mentioned above, access to new biomarkers or imaging technologies might not be given for many health service providers. Even more, combining new technologies and biomarkers will increase the cost per patient screened which might not be sustainable in many healthcare systems. Clearly, these factors must be considered as limitations for the implementation of advances that could potentially lead to improved delivery of lung cancer screening.

The following limitations in our review should be kept in mind when interpreting our findings. First, this review was not performed systematically, may not be comprehensive, and as such might be subject to selection bias. Second, the strength of evidence provided by the studies presented in this review varies due to study designs. Therefore, a direct comparison of diagnostic tools and prognostic models and, in particular, the selection of most promising approaches would be challenging if not impossible. However, our main intention was to raise awareness by presenting the current research on LC screening, promoting both further research and technological progress.

## 7. Conclusions

To conclude, this review aimed at providing a holistic view of how the entirety of LC screening related aspects might be advanced in the near future. An advanced selection approach incorporating additional risk factors and biomarkers might improve the selection of candidates eligible for subsequent LDCT screening. At screening, image reading efficacy and accuracy might be augmented by IT tools, helping radiologists to cope with the growing workload resulting from LC screening programs. At the post-CT management, semi-automatic volume measurements potentially increase the precision and predictive value of follow-up PN imaging, and an integrative approach involving clinical parameters, radiomics, and biomarkers might optimize the characterization and management of CT-detected PN. Further large-scale validation studies are obviously required, but the integration of the scientific and technological progress into LDCT-based LC screening programs has the potential to clearly improve the performance of LC screening generally.

## Figures and Tables

**Figure 1 cancers-15-01218-f001:**
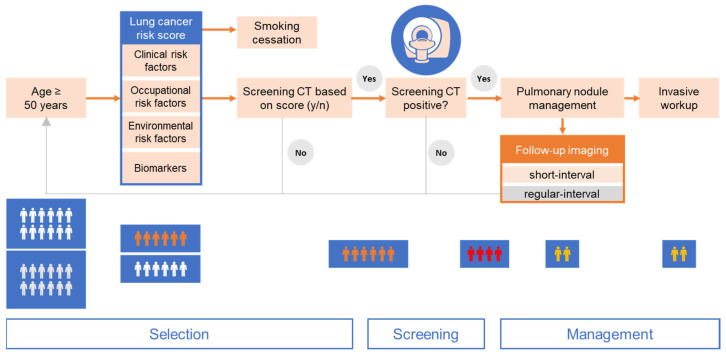
Three-step strategy to overcome current limitations in lung cancer screening by computed tomography (CT). Selection: Improved criteria basing on clinical, occupational, and environmental risk factors as well as biomarkers could broaden the eligible population and increase pre-test probability (age threshold according to current USPSTF criteria [24]). Screening: Technical progress enables the automation of nodule identification and reduction of false positives. Management: Novel criteria to assess the probability of malignancy allow safer decisions on further nodule management and follow-up screening intervals (pulmonary nodule management is depicted in more detail in Figure 2).

**Figure 2 cancers-15-01218-f002:**
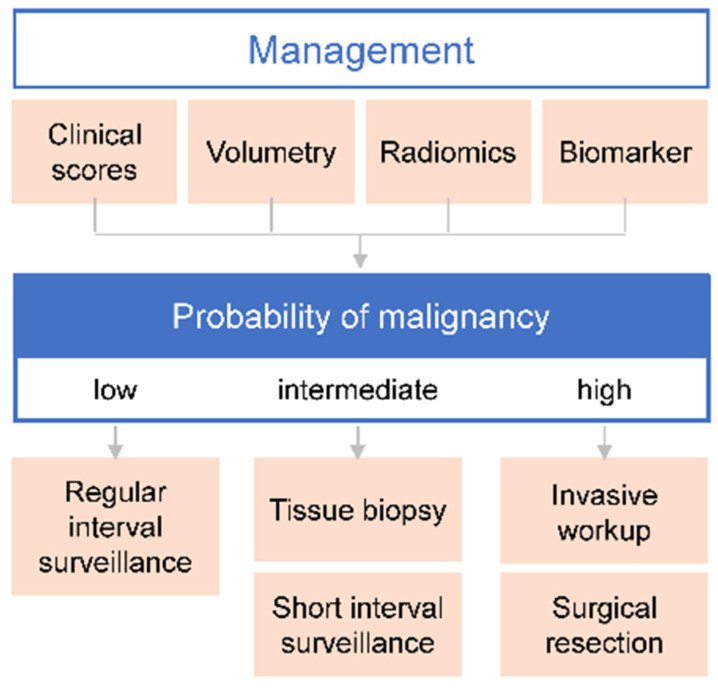
Management of pulmonary nodules after low dose computed tomography scanning.

## Data Availability

Not applicable.

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
