# Peer review of "Clinical Scores, Biomarkers and IT Tools in Lung Cancer Screening—Can an Integrated Approach Overcome Current Challenges?"

_cancers, 2023, doi:10.3390/cancers15041218_

Round 1
Reviewer 1 Report
In this scoping review article, the authors describe the limitations and advances of the three different phases of the lung cancer screening (LCS) process: the patient selection/eligibility phase, technical aspects of the screening procedure, and evaluation and management of CT-detected pulmonary nodules (PNs). The breadth and depth of the cited manuscripts was appropriate for the noted goal of the manuscript, and overall the review was comprehensive and well structured. However, as noted in specific comments below, additional attention and revison is needed with respect to some of the authors’ declarative statements and noted citations. Moreover, the addition of 1, or preferable 2 tables, that summarize the noted literature, could enhance both the framing and the readability of the manuscript.
Lastly, the issue that was not addressed in this scoping review, and perhaps should be explicitly addressed within the paper or noted as a limitation, is feasibility - both in the U.S. and internationally. Specifically, how feasible or easy would it be for clinicians, health systems, or national health service providers to implement the recommended options or advances that could lead to more optimal delivery of LCS?
Specific comments/issues:
1. Line 48, reference # 4 is not consistent or applicable to the noted statement.
2. Line 62, reference #12 does not support the statement associated with anxiety.
3. Lines 64-67, the noted citations are associate with the use of spirometry to detect COPD. They are not germane to “dedicated IT solutions” and associated declarative statements.
4. Lines 134-135: What does "apparently" mean in this context. Was there significant evidence noted in the cited literature to support the statement that PLCOm2012 produces superior performance metrics relative to USPSTF 21?
5. Lines 140-143: The authors' statement was similar to a noted statement in the abstract of reference 28 - which was also a review article. However, in contrast to this statement, the authors do not directly link published evidence regarding the attributable risk of lung cancer in never-smokers - particularly in Asian females.
6. Lines 259-268: describe well known and frequently noted incidental findings associated with LCS. Consistent with the goal of the manuscript to provide or improve integrated approach to LCS the paragraph should consider recommendations for how to approach or care for patients with Ifs.
7. Overall Section 3. Selection Phase: Given this is a scoping review, the authors should consider adding a table associated with the Selection phase that notes the additional demographic, environmental, and clinical/biomarker issues that should be considered (in one column), with a second column noting the reference and/or source of the evidence.
8. The statement and noted reference on lines 308 and 309 is troubling and should be revised or removed. First, the citation has nothing to do with volumetric measurement of PNs. Second, the evidence cited is based on simulation model of 2013 NCD and SEER data - not direct or patient level evidence from LCS programs using volumetric measurement, or other.
9. Sections 3.2 and 5.4 both provide extensive details on the various biomarkers, some overlapping, along with their associated performance metrics, for use in either the screening or PN management phases. These sections could either be consolidated and/or delineated noting the evidence supporting which biomarker assays are best for consideration in the context of either screening OR PN management. Alternatively, a Table should be included listing the various available biomarkers used for screening vs PN management, along with their noted performance metrics, and citations.
Author Response
The authors would like to thank reviewer # 1 for the detailed and careful review as well as the valuable suggestions. In the following we are happy to reply to the specific and general comments point by point:
- Line 48, reference #4: we agree to the reviewer’s comment and replaced reference # 4 to fully support our statements.
- Line 62, reference #6: we agree to the reviewer’s comments and added further references to fully support our statement related to anxiety.
- Lines 64-67, we agree to the reviewer’s comments and replaced references to fully support our statements made in this section.
- Lines 134-35, we agree to the reviewer’s comments and changed the language avoid misunderstandings as pointed out by the reviewer.
- Lines 140-43, we agree to the reviewer’s comments and changed the language to avoid reference errors with the previous sentence.
- Lines 259-268, we agree to the reviewer’s comments and thank for this valuable suggestion. We added a short section to discuss the work up and medical consequences of incidental findings.
- See 9.
- Lines 308-309. We agree to the reviewer’s comment and decided to delete the respective sentence.
- We do understand the reviewer’s point of view. When drafting our scoping review we deliberately decided against summary tables for the following reasons: 1) we intended to highlight the value which might come from the integration of different clinical or lab-biomarkers etc. in different phases of a CT based lung cancer screening program. We did not intend to create a systematic literature review as this would by far overload the paper given the dimension of existing evidence. Therefore, we decided to stick to a narrative style in our manuscript 2) summarizing papers in a table format on the other side would imply a level of completeness of the literature search which is not existing in our work nor was it intended. Therefore, we decided not to follow the reviewer’s suggestions at this particular point
- Last comment, we thank the reviewer for this valuable and thoughtful comment and are happy to address it in our manuscript. We added an entire section (lines 537-42 in the revised manuscript) in the discussion section to add this very important perspective into our work.

Reviewer 2 Report
The authors review current advances in the screening and malignancy probability assessment of CT-detected pulmonary nodules, and this study provides a comprehensive approach to assessing the probability of PN malignancy, predicting whether they are benign or malignant and determining appropriate further procedures.
Minor comments:Article validation is basically with AUC, specificity and sensitivity, whether more external data can be added to validate.
Author Response
The authors would like to thank reviewer # 2 for the detailed and careful review as well as the valuable minor comment given.
Round 2
Reviewer 1 Report
Appropriate and sufficient responses by the authors to the previous review.